# Probiotics, the Immune Response and Acute Appendicitis: A Review

**DOI:** 10.3390/vaccines11071170

**Published:** 2023-06-28

**Authors:** Carmine Petruzziello, Angela Saviano, Veronica Ojetti

**Affiliations:** 1Emergency Department, Ospedale San Carlo di Nancy—GVM Care & Research, 00165 Rome, Italy; carminepetruzziello@live.it; 2Emergency Department, Ospedale Policlinico A. Gemelli, 00135 Rome, Italy; angela.saviano@policlinicogemelli.it; 3Università Cattolica del Sacro Cuore, 00135 Rome, Italy

**Keywords:** acute appendicitis, gut, probiotics, emergency department, *lactobacilli*, bifidobacteria, immune system

## Abstract

Acute appendicitis is a common reason for admission to the Emergency Department (ED). It affects almost 70% of people under 30 years of age and 10% over 60 years of age. Its diagnosis includes the combination of clinical signs, laboratory tests and imaging. For years, surgical appendectomy has been the first-line therapy for acute appendicitis, but currently the management has shown some changes, in particular in patients with uncomplicated appendicitis. Recent studies have investigated the use of probiotics as an adjunctive therapy with promising results in conferring health benefits to patients with acute appendicitis. The aim of our review is to summarize the results of clinical studies about probiotics and the immunological response in acute appendicitis, discussing the limitations and future directions of this research.

## 1. Introduction

Acute appendicitis is a common surgical emergency that affects a significant portion of the general population, with an estimated incidence of 7–8% [1]. It is characterized by inflammation and infection of the appendix, a small organ located at the junction of the small and large intestines. Traditionally, the mainstay of treatment for acute appendicitis has been surgical removal of the inflamed appendix, known as an appendectomy. However, in recent years, there has been a growing interest in exploring alternative approaches for the medical management of this condition, partly driven by concerns over post-operative complications and the associated costs of surgery.

One approach that has gained attention is the use of antibiotics as the primary treatment for selected patients with uncomplicated acute appendicitis, including pregnant individuals. A recent randomized controlled trial (RCT) has demonstrated the efficacy of an antibiotic-first approach in these cases, offering a potential non-surgical alternative [2]. The WSES guidelines recommend the following: amoxicillin/clavulanate 1.2–2.2 g 6-hourly or ceftriaxone 2 g 24-hourly + metronidazole 500 mg 6-hourly or cefotaxime 2 g 8-hourly + metronidazole 500 mg 6-hourly [2]. Furthermore, a randomized controlled trial conducted by Park et al. showed that uncomplicated acute appendicitis may safely resolve spontaneously with rest and fluids, resulting in similar treatment failure rates, shorter hospital stays and reduced costs compared to antibiotic treatment [3].

However, in recent years, researchers have begun exploring additional treatment strategies to optimize outcomes and further improve patient care. Among these strategies is the potential use of probiotics as an adjunctive therapy in the management of acute appendicitis. Probiotics are live microorganisms that, when administered in adequate amounts, confer health benefits to the host. They have been extensively studied for their ability to modulate the gut microbiota, enhance immune function and promote overall well-being.

In the context of acute appendicitis, probiotics have the potential to offer several advantages. Firstly, they may help reduce inflammation within the appendix, mitigating the severity of the condition. Secondly, probiotics have been shown to improve gut barrier function, which could aid in preventing the spread of bacteria from the inflamed appendix to other parts of the abdominal cavity [4]. This is crucial, as bacterial translocation can lead to complications, such as peritonitis. Additionally, probiotics may have a positive impact on the overall microbial balance in the gut, promoting a healthier environment and potentially influencing the course of appendicular inflammation.

Several studies have investigated the use of probiotics in the management of gastrointestinal diseases, yielding promising results [4,5]. These studies have explored different probiotic strains, dosages and treatment durations, aiming to determine the most effective approach. In this review, we will delve into the pathophysiology of acute appendicitis, shedding light on the mechanisms underlying this condition. Furthermore, we will explore the role of the gut microbiota in acute appendicitis and discuss how probiotics may exert their beneficial effects. Moreover, we will summarize the findings of clinical studies investigating the use of probiotics as an adjunctive therapy in acute appendicitis management, highlighting their potential benefits and limitations. Finally, we will address the future directions of this research, emphasizing the need for the further exploration and elucidation of the optimal use of probiotics in the context of acute appendicitis management.

## 2. Pathophysiology of Acute Appendicitis

The exact cause of acute appendicitis is not fully understood, but it is thought to result from a combination of factors, including luminal obstruction by fecaliths, bacterial overgrowth, inflammation, infectious agents, dietary, genetic and hygienic factors, ischemia and traumatic causes. The appendix is a small, blind-ended tube connected to the cecum, which is part of the large intestine [1,2]. Luminal obstruction occurs when the opening of the appendix becomes blocked, usually due to the presence of a fecalith (a hardened mass of stool) or lymphoid hyperplasia (enlargement of lymphoid tissue). This obstruction leads to the distension of the appendix and impairs the normal drainage of its contents [2,3].

The appendix is colonized by various bacteria, with *Escherichia coli* and bacteria belonging to the *Bacteroides fragilis* group being the most commonly involved pathogens. When luminal obstruction occurs, it creates an environment conducive to bacterial overgrowth and colonization within the appendix. The stagnant contents and increased bacterial load can trigger an inflammatory response, leading to infection and inflammation of the appendix.

The inflammatory process in acute appendicitis involves the release of cytokines, chemokines and other pro-inflammatory mediators. These signaling molecules attract immune cells, such as neutrophils and macrophages, to the site of inflammation.

Literature studies [6] reveal a substantial increase in inflammatory markers associated with Th1 and Th17 cell responses. In particular, IL-17 in patients with appendicitis, leads to the recruitment of neutrophils to the site of infection with the production of cytokines and chemokines. Many studies underline the relevance of IL-17 in activating the immune response in patients affected by appendiceal inflammation. It has been demonstrated that IL-17 is particularly expressed during infection by *Escherichia coli*, which represents the most common detected bacterium in acute complicated appendicitis. The pathway IL-23/IL-17 associated with Th17, with high production of IL-27, IL-17A and IL-23A, has been described in gangrenous appendicitis.

Furthermore, immunological studies revealed the overexpression of mRNA of proteins produced by T cells and of co-stimulatory CD40l. This last factor has been considered an independent predictor of acute appendicitis. Moreover, the subunits (alpha and beta subtypes) of proteins produced by T cells are overexpressed in patients with phlegmonous appendicitis [6], thus suggesting the theory of viral infections as an etiological factor.

The recruited immune cells attempt to eliminate the bacterial infection; however, their presence and the release of inflammatory substances contribute to tissue damage and further amplify the inflammatory response.

If left untreated, acute appendicitis can progress to complications, such as appendix rupture, abscess formation, and peritonitis. Appendix rupture occurs when the increased pressure within the appendix exceeds its capacity, causing the organ to burst. This leads to the release of infected contents into the abdominal cavity, increasing the risk of peritonitis—inflammation of the lining of the abdominal cavity. Abscess formation may also occur when the body’s immune response walls off the infected appendix, creating a pocket of pus.

The severity of acute appendicitis can vary from mild cases with localized inflammation to more severe cases with extensive infection and tissue damage. Factors, such as the degree of luminal obstruction, the virulence of the infecting bacteria and the individual’s immune response, all contribute to the course and outcome of the condition.

The Alvarado score is a 10-point clinical score for the diagnosis of acute appendicitis. It is based on symptoms, signs and diagnostic tests [7]. Another score is the appendicitis inflammatory response (AIR) score, which outperforms the Alvarado score. It introduces C-reactive protein (CRP) into the score and is able to predict acute appendicitis with higher specificity, decreasing the rate of negative appendicectomy [8].

Understanding the pathophysiology of acute appendicitis is crucial for developing effective treatment strategies. By targeting the underlying mechanisms, interventions can aim to alleviate luminal obstruction, control bacterial overgrowth and modulate the inflammatory response.

To date, the management of acute appendicitis is based on the 2020-WSES guidelines that recommended the non-operative management in cases of uncomplicated acute appendicitis, using antibiotics as the first approach. The success of this approach requires the careful selection of patients and exclusion of patients with gangrenous acute appendicitis, abscesses and diffuse peritonitis [9]. Initially, antibiotics can be intravenous with subsequent conversion to oral antibiotics. Regarding complicated acute appendicitis, some studies support initial antibiotics with delayed operation, whereas others support immediate operation (usually, laparoscopic appendicectomy). In complicated acute appendicitis, the choice of intravenous antibiotics must be effective against enteric Gram-negative organisms and anaerobes, including *Escherichia coli* and *Bacteroides* species, and it should start as soon as the diagnosis is established. Broader-spectrum antibiotics can include piperacillin-tazobactam, ticarcillin-clavulanate, ampicillin-sulbactam or imipenem-cilastatin. Regarding perforated acute appendicitis, the most common combination is ampicillin, metronidazole (or clindamycin) and gentamicin. Alternatives include ceftriaxone-metronidazole or ticarcillin-clavulanate plus gentamicin, based on the epidemiology of bacteria [9].

The exploration of adjunctive therapies, such as probiotics, seeks to optimize these treatment approaches, avoiding inappropriate surgical measures, and improve outcomes for patients with acute appendicitis.

## 3. Role of the Gut Microbiota in Acute Appendicitis

The gut microbiota, consisting of trillions of microorganisms that inhabit the gastrointestinal tract, plays a crucial role in maintaining gut homeostasis and immune functions.

Similar to the colon, the appendix is inhabited by *Firmicutes*, *Bacteroidetes*, *Actinobacteria* and *Proteobacteria* species. Appendicitis is associated with an altered gut microbiota. Interestingly, in the surgically removed appendices of patients with acute appendicitis, some oral pathogens and some pathogens of the distal gut, such as *Fusobacterium* and *Parvimona*, have been identified [10].

As described before, literature studies have demonstrated differences in the intestinal microbiota of patients with acute appendicitis and healthy populations. In particular, the gut microbiota of patients with acute appendicitis is rich in *Fusobacteria* species. However, the impact of this diversity on the severity of the disease has not been clarified yet. Furthermore, diversity of anaerobes in the appendix compared to healthy controls has also been described.

Dysbiosis, which refers to an imbalance of the composition and diversity of the gut microbiota, has been implicated in the pathogenesis of various gastrointestinal disorders, including inflammatory bowel disease, irritable bowel syndrome and colorectal cancer. In the case of acute appendicitis, dysbiosis may also play a role in the initiation and progression of inflammation and infection [11,12].

The luminal obstruction that leads to the development of acute appendicitis can disrupt the normal balance of the gut microbiota. The obstruction prevents the normal flow of intestinal contents, causing stagnation and altering the microenvironment within the appendix. This disruption creates an opportunity for pathogenic bacteria to overgrow and colonize the appendix [13].

Studies have shown that certain bacteria, such as *Clostridium* and *Prevotella*, are more abundant in individuals with acute appendicitis than in healthy controls, while other beneficial bacteria, like *Streptococcus*, are reduced [11]. *Prevotella*, an opportunistic pathogen commonly found in the gut microbiota, has been associated with systemic infections and could potentially contribute to the alteration of the gut barrier function, thereby playing a critical role in the pathogenesis of acute appendicitis. The overgrowth of *Prevotella* and other pathogenic bacteria in the appendix may trigger an inflammatory response, leading to the release of pro-inflammatory mediators and the activation of the immune system [11]. This exacerbates the inflammatory cascade, resulting in tissue damage and worsening of the condition.

Furthermore, the appendix has the highest concentration of gut-associated lymphoid tissue (GALT) in the intestine, indicating its importance in the immune response within the gastrointestinal tract. Changes in the gut microbiota can directly influence the GALT in the appendix, potentially inducing an increase in B cell levels and altering immune functions [11]. The interaction between the gut microbiota and the appendix immune system highlights the potential impact of dysbiosis on the development and progression of acute appendicitis.

Interestingly, a study by Cai et al. [10] showed that after appendicectomy, there is an alteration in the bacterial and fungal composition of the gut microbiota. In their study, the authors found that the alpha diversity did not change, while the beta diversity revealed an alteration of the gut bacterial composition. In their cross-sectional study, the authors investigated the association between appendectomy and the gut microbiota composition using fecal samples of 30 healthy individuals with prior appendectomy and 30 healthy individuals without appendectomy. Cai at al. [10] reported a modified composition of gut bacteria with regards to Firmicutes, Bacteroidetes, Proteobacteria and Fusobacteria, with a change in the prevalence of *Butyricimonas*, *Roseburia*, *Butyricicoccus*, *Barnesiella* and *Odoribacter* after appendicectomy. Importantly, these “more abundant” gut bacteria belonged to the group of short-chain fatty acids (SCFAs) producers. SCFAs, such as butyric acid, propionic acid and acetic acid play key roles in the regulation of immunity, in the protection of intestinal mucosa and also in the protection against inflammation and in the provision of epithelial cell energy. The authors concluded that appendicectomy marked a huge impact on gut bacteria and fungi, which was particularly robust for fungi (up to five years), and with significant changes in the complex fungal–bacterial and fungal–fungal interactions. Additional studies are needed to better deepen this topic.

Another study by Shi et al. [14] identified seven bacteria in the gut microbiota after appendicectomy (*B. vulgatus*, *B. fragilis*, *P. ruminicola*, *Veillonella dispar*, *P. dentalis*, *P. fusca*, *P. denticola*) involved in tumorigenesis and bowel inflammation. These bacteria can induce DNA damage and activated Th17 cell immune responses. Moreover, they cause leucocyte chemotaxis and stimulate the production of pro-inflammatory cytokines. Moreover, *Veillonella* was reported to be associated with inflammatory bowel diseases due to cytokine induction and activation of the oncogenic p38 MAPK pathway.

Understanding the role of the gut microbiota in acute appendicitis provides insights into the complex interplay among the gut microbiota, inflammation and infection. It emphasizes the importance of restoring a healthy microbial balance in the gut as a potential therapeutic approach for managing acute appendicitis. By targeting dysbiosis through interventions, such as probiotic administration, it may be possible to modulate the gut microbiota, reduce inflammation and promote a favorable environment for appendix recovery.

In the next section, we will explore the potential benefits of probiotics in the management of acute appendicitis and discuss the findings of clinical studies investigating their use as an adjunctive therapy.

## 4. Methodology

This review included studies published in any language in the last 20 years about the role of gut microbiota, its modulation with probiotics and gastrointestinal diseases, focusing on acute appendicitis. We searched clinical trials, systematic reviews and observational studies (case-control, case series, longitudinal and cross-sectional).

We extracted data using parameters such as the following: title, period of research, type of study, abstract. We searched PubMed^®^, Web of Science^®^, Up-to-Date^®^ and Cochrane^®^. No ethical approval was needed to perform this review. The principal items that we searched were as follows: microbiota AND gastrointestinal diseases, probiotics AND appendicitis AND/OR immune response, uncomplicated acute appendicitis AND probiotics, complicated acute appendicitis AND probiotics, gut microbiota AND/OR appendix, dysbiosis AND gut disorders AND/OR acute appendicitis.

## 5. Potential Benefits of Probiotics on Acute Appendicitis

Probiotics have shown significant potential in the management of acute appendicitis, offering various benefits that can aid in the treatment and prevention of complications [15]. One of the primary advantages of probiotics is their ability to modulate the gut microbiota and restore the microbial balance [16]. By introducing beneficial bacteria into the digestive system, probiotics can help suppress the growth and colonization of pathogenic microorganisms, reducing the risk of inflammation and infection [17,18]. Furthermore, in cases where inflammation is already present, probiotics have been shown to attenuate mucosal damage by limiting bacterial translocation [19,20].

The scientist Elie Metchnikoff introduced the concept of probiotics, publishing a report in which he underlined the “longevity” of Bulgarians who consumed fermented milk products containing *lactobacilli* strains [21]. Since then, probiotics had been widely produced and consumed, due to the many proprieties described, such as the ability to manipulate intestinal microbial communities, to suppress pathogens, to stimulate epithelial cell proliferation and promote the differentiation and the fortification of the intestinal barrier. Furthermore, probiotics may induce the production of β-defensin and IgA. They can maintain the tight junctions and induce the production of mucus. Literature studies demonstrate that probiotics can mediate immunomodulation and cytokine secretion acting on NFκB and MAPKs pathways, which are the same involved in the proliferation and differentiation of immune cells (for example, T cells) or epithelial cells. Moreover, probiotics improve gut motility and nociception, and they can regulate the expression of pain receptors and the secretion of neurotransmitters [21]. Therefore, they can induce benefits in cases of acute appendicitis [22].

Research has demonstrated that probiotics can decrease the levels of pro-inflammatory cytokines and chemokines, such as tumor necrosis factor-alpha (TNF-alpha) and interleukin-6 (IL-6), which are typically elevated in acute appendicitis [16]. By enhancing the production of mucus and strengthening the tight junctions between intestinal cells, probiotics contribute to a more robust gut barrier that prevents the translocation of bacteria and harmful substances from the gut lumen into the bloodstream [19,23,24]. Additionally, probiotics can reduce the activation of Toll-like receptor immune cells while promoting the induction of T-regulatory (T-reg) cells, thereby restoring gut homeostasis [19]. Furthermore, probiotics exhibit the potential to improve immune functions and reduce the risk of complications in acute appendicitis [24]. The immune system plays a vital role in responding to infections and inflammation, and probiotics have been shown to enhance immune functions by increasing the production of antibodies and activating key immune cells, such as natural killer cells and T cells. The interaction between the gut microbiota and the human immune system is crucial for maintaining immune quiescence in the gut, and probiotics play a significant role in this balance [17].

Studies conducted by Ojetti et al. [25] have provided insights into the efficacy of probiotic supplementation in the treatment of acute uncomplicated diverticulitis (AUD), which shares similarities to acute appendicitis in terms of inflammation and obstruction. In patients with AUD, supplementation with an anti-inflammatory strain of *Lactobacillus reuteri* 4659, along with bowel rest and fluids, resulted in a significant reduction in abdominal pain, blood inflammatory markers (C-reactive protein) and fecal inflammatory markers (calprotectin) compared to observations in the placebo group.

Similarly, another study [19] demonstrated the effectiveness of probiotic supplementation (a mix of *Bifidobacterium lactis LA 304*, *Lactobacillus salivarius LA 302* and *Lactobacillus acidophilus LA 201*) with anti-inflammatory properties during antibiotic treatment for AUD. This probiotic mixture effectively reduced abdominal pain, inflammation and the duration of hospitalization. Notably, *L. salivarius Ls33* and *L. acidophilus* were among the top-performing probiotics, inducing high levels of anti-inflammatory interleukin-10 (IL-10) while minimizing the production of pro-inflammatory interleukin-12 (IL-12) [19].

These studies provide compelling evidence for the potential benefits of probiotics in the management of acute appendicitis. By modulating the gut microbiota, enhancing the gut barrier function and improving the immune response, probiotics offer a promising adjunctive therapy that may reduce inflammation, prevent complications and contribute to better patient outcomes. However, further research is needed to optimize probiotic strains, dosages and treatment protocols specific to acute appendicitis and to determine the long-term effects.

## 6. Use of Probiotics in Acute Appendicitis: An Immunological Perspective

The use of probiotics in acute appendicitis offers promising potential from an immunological perspective. Numerous studies have demonstrated that probiotics can regulate both the innate and adaptive immune systems, exerting their effects on various immune cells, including dendritic cells, B and T lymphocytes and macrophages [19]. For instance, research conducted by Maldonado et al. investigated the influence of *Lactobacillus casei CRL 431* and *Lactobacillus paracasei CNCM I-1518* on the immune system and found that probiotics can activate toll-like receptors and stimulate the immune response [19].

Probiotics have been shown to increase the production of anti-inflammatory cytokines and chemokines while interacting with intestinal epithelial cells and attracting macrophages and mononuclear cells. Certain strains of probiotics can activate regulatory immune T cells, which release anti-inflammatory interleukin (IL)-10. Interestingly, probiotics also reinforce the intestinal barrier by enhancing the expression of tight junction proteins, mucins, Paneth cells and Goblet cells, thus promoting gut health [25]. Moreover, beneficial bacteria, such as *Oscillibacter* and *Prevotella*, produce anti-inflammatory metabolites that support the differentiation of anti-inflammatory Treg (Type 1 regulatory T) cells in the human gut [26].

Another notable probiotic strain, *Lactobacillus rhamnosus Lcr35*, has been found to induce changes in gene expression, leading to an increase in the production of Th1/Th17 cytokines, such as IL-1, TNF and IL-10. Additionally, it stimulates the maturation of dendritic cell membranes by upregulating HLA-DR and TLR4, and it exerts a dose-dependent immunomodulatory effect on dendritic cells [27]. Studies focusing on *Lactobacillus acidophilus NCFM* and *Lactobacillus salivarius Ls-33* have demonstrated their protective effects against colitis in mice, partly attributed to the influence of these probiotics on Tregs (regulatory T cells) [28]. Furthermore, in mice receiving these probiotics, a Treg-favorable environment was observed [29]. Other research has shown that a probiotic mixture containing *Lactobacillus acidophilus*, *Lactobacillus reuteri*, *Lactobacillus casei* and *Bifidobacterium bifidum* stimulates dendritic cells to produce IL-10 and TGF-β, promoting the development of regulatory T cells (Tregs) [28]. Moreover, this probiotic combination downregulated the production of T helper (Th) 1, Th2 and Th17 cytokines, which are associated with inflammation [20].

In summary, probiotics play a significant role in maintaining the balance between innate and adaptive immune responses through their direct and indirect interactions with epithelial cells, dendritic cells and B and T lymphocytes [26]. Additionally, probiotics can regulate gene expression and mediate immunomodulation by targeting specific signaling pathways [28] (Figure 1). These immunological effects of probiotics highlight their potential as a valuable therapeutic approach in acute appendicitis, as they can modulate immune responses, promote anti-inflammatory effects and contribute to the overall management of the condition (Figure 2). However, further research is still necessary to fully understand the specific mechanisms and optimal strains and dosages of probiotics in the context of acute appendicitis.

## 7. Clinical Studies on Probiotics in Acute Appendicitis

Several clinical studies have been conducted to investigate the efficacy of probiotics for the management of gastrointestinal diseases [30] with a possible role also in the context of acute appendicitis.

One notable randomized controlled trial, published in 2013, focused on the use of *Lactobacillus plantarum* in patients with acute uncomplicated appendicitis [31]. The study included 60 patients who were randomly assigned to receive either *L. plantarum* or a placebo for three days prior to surgery. The results demonstrated that the administration of *L. plantarum* led to a significant reduction in postoperative complications, including wound infection, ileus (intestinal obstruction) and fever. Furthermore, the group receiving *L. plantarum* exhibited a significantly shorter duration of hospital stay compared to the placebo group [31].

A study by Tang et al. [32] revealed that during abdominal infections, some substances, such as H2O2, bacteriocins and biosurfactant secreted by gut bacteria, such as *Lactobacillus*, can kill pathogenic microorganisms and better control the inflammatory response, enhancing the therapeutic effects. The metabolites produced by probiotic strains play a key role in the regulation of host–disease interactions. For example, *Lactobacillus plantarum BGCG11* produces an extracellular polysaccharide (EPSCG11) that is able to alleviate inflammatory pain in animal models (rats), reduce the expression of proinflammatory molecules and increase anti-inflammatory factors (such as IL-10 and IL-6), with a potent anti-edema and anti-hyperalgesic effects. Therefore, probiotics can provide important medical results, even if the specific strains or specific prebiotics in patients with acute appendicitis have not been determined yet.

These clinical studies provide valuable insights into the potential benefits of probiotics in acute appendicitis. The results consistently indicate that probiotic supplementation can contribute to better outcomes, including a reduction in postoperative complications and shorter hospital stays. By modulating the gut microbiota, enhancing immune function and improving the gut barrier integrity, probiotics offer a promising adjunctive therapy for the management of acute appendicitis. However, further research is still needed to determine the optimal strains, dosages and duration of probiotic supplementation, as well as to evaluate long-term outcomes and potential adverse effects.

## 8. Limitations and Future Directions

Despite the promising results of clinical studies investigating the use of probiotics in acute appendicitis, several limitations and challenges must be addressed before probiotics can be widely recommended in clinical practice. First, there is a lack of standardization in the strains, doses and duration of probiotic supplementation used in clinical studies. This makes it difficult to compare the results of different studies and to determine the optimal use of probiotics in acute appendicitis. Second, the mechanisms by which probiotics exert their beneficial effects in acute appendicitis are not fully understood. Further research is needed to elucidate the underlying mechanisms and to identify the most effective probiotic strains and doses. Third, larger and more diverse patient populations need to be studied to determine the generalizability of these findings. Most of the studies to date have been conducted on relatively small and homogenous populations, and more research is needed to determine the efficacy and safety of probiotics in a wider range of patients.

## 9. Conclusions

Probiotics may be a useful adjunctive therapy for the management of acute appendicitis, acting both on inflammation and on the regulation of the innate and adaptative immune systems. Probiotics have been shown to reduce postoperative complications and improve recovery after surgery. However, further research is needed to determine the optimal use of probiotics in this setting, including the most effective strains, doses and duration of supplementation. Despite these challenges, the use of probiotics for acute appendicitis represents a promising area of research that may ultimately lead to improved outcomes.

## Figures and Tables

**Figure 1 vaccines-11-01170-f001:**
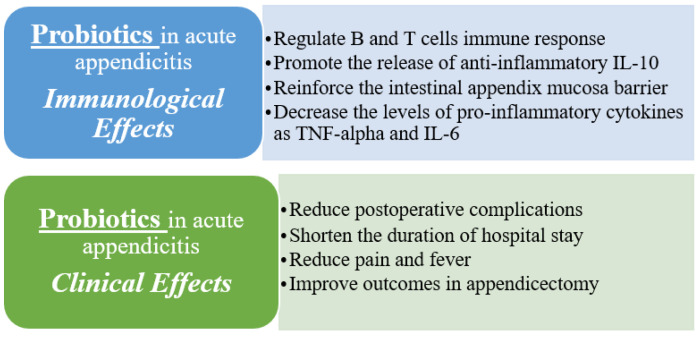
Effects of probiotics on acute appendicitis.

**Figure 2 vaccines-11-01170-f002:**
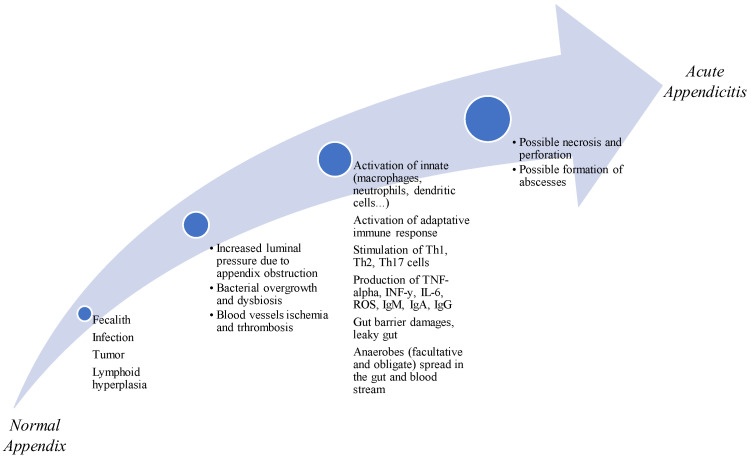
Inflammation pathway in acute appendicitis.

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
