# Peer review of "Probiotics, the Immune Response and Acute Appendicitis: A Review"

_vaccines, 2023, doi:10.3390/vaccines11071170_

Round 1
Reviewer 1 Report
it is an interesting review article about the use of probiotics in acute appendicitis and i think it adds in the literature according to this direction.
In this research we read a review of the already written articles about the use o probiotics in acute appendicitis and the benefits after their usage in postoperative complications of appendicitis there are not other review articles as i know in this field so of course this paper adds to the literature.The references are appropriate. As a review of the control trials that are already exist in this field of course it adds to the literature. The conclusions and the limitations of the study are appropriate. The only think that i have to mention is that the authors should refer to the methodology of the article search in order to conclude to this results.
Author Response
The only think that i have to mention is that the authors should refer to the methodology of the article search in order to conclude to this results.
Thank you so much for your observations,
we added the methodology as you suggested
Reviewer 2 Report
This manuscript is well organized and interesting. However, there are some concerns.
Major points
Section 6 described about clinical studies of prebiotics in acute appendicitis. The references presented by the authors did not match what was stated in the manuscript. The reference No.33 do not deal with acute appendicitis. The study protocol in the reference No.34 do not use probiotics, and the number of the patients was different. The author of the reference No.35 do not include Tang et al, and the statements described in it is different from the statements in Lines 330-340. The references from No.36 to No.40 do not deal with probiotics. The reference No.41 was not indicated in the manuscript. The reference No.42 does not deal with the efficacy of probiotics therapy for patients with acute appendicitis following surgery.
Minor points
The reference No.2 is not meta-analysis. It is RCT.
The statement line.32-35 need a reference.
The reference of line.35-38 is probably the reference No.2.
Author Response
Major points
Section 6 described about clinical studies of prebiotics in acute appendicitis. The references presented by the authors did not match what was stated in the manuscript. The reference No.33 do not deal with acute appendicitis. The study protocol in the reference No.34 do not use probiotics, and the number of the patients was different. The author of the reference No.35 do not include Tang et al, and the statements described in it is different from the statements in Lines 330-340. The references from No.36 to No.40 do not deal with probiotics. The reference No.41 was not indicated in the manuscript. The reference No.42 does not deal with the efficacy of probiotics therapy for patients with acute appendicitis following surgery.
Thank you so much for your observations, we checked and modified all these references
Minor points
The reference No.2 is not meta-analysis. It is RCT. We modified
The statement line.32-35 need a reference. We added references
The reference of line.35-38 is probably the reference No.2. We modified reference
Reviewer 3 Report
Acute appendicitis is a common reason of admission to the Emergency Department (ED). It affects almost 70% of people under 30 years old and 10% over 60 years old. Its diagnosis includes the combination of clinical signs, laboratory tests and imaging. For years, surgical appendectomy has been the first-line therapy for acute appendicitis, but currently the management has shown some changes, in particular in patients with uncomplicated appendicitis. Recent studies have investigated the use of probiotics as an adjunctive therapy with promising results in conferring health benefits in patients with acute appendicitis. The aim of our review is to summarize the results of clinical studies about probiotics, immunological response in acute appendicitis, discussing the limitations and future directions of this research.
It is interesting topic and the result is good. However, there still have some other issue need to check.
1. In the introduction part. Probiotics should be added reference, which also include the prebiotic in diet (Critical Reviews in Food Science and Nutrition. Doi: 10.1080/10408398.2021.1995842.).
2. The inflammation pathway graph should be added.
Please update the reference in recent years.
Acute appendicitis is a common reason of admission to the Emergency Department (ED). It affects almost 70% of people under 30 years old and 10% over 60 years old. Its diagnosis includes the combination of clinical signs, laboratory tests and imaging. For years, surgical appendectomy has been the first-line therapy for acute appendicitis, but currently the management has shown some changes, in particular in patients with uncomplicated appendicitis. Recent studies have investigated the use of probiotics as an adjunctive therapy with promising results in conferring health benefits in patients with acute appendicitis. The aim of our review is to summarize the results of clinical studies about probiotics, immunological response in acute appendicitis, discussing the limitations and future directions of this research.
It is interesting topic and the result is good. However, there still have some other issue need to check.
1. In the introduction part. Probiotics should be added reference, which also include the prebiotic in diet (Critical Reviews in Food Science and Nutrition. Doi: 10.1080/10408398.2021.1995842.).
2. The inflammation pathway graph should be added.
Please update the reference in recent years.
Author Response
In the introduction part. Probiotics should be added reference, which also include the prebiotic in diet (Critical Reviews in Food Science and Nutrition. Doi: 10.1080/10408398.2021.1995842.)
We added this reference as you suggested
The inflammation pathway graph should be added.
We added a figure with inflammation pathway as you suggested
Please update the reference in recent years.
We checked and modified references as you suggested
Round 2
Reviewer 2 Report
The authors have made appropriate corrections. Good job.
Author Response
Thank you so much
Reviewer 3 Report
The author has responsed the reviewer's comment point by point. However, the reference should be update with corret page and Doi.
The author has responsed the reviewer's comment point by point. However, the reference should be update with corret page and Doi.
Author Response
We modified references as suggested